# Contribution to the 3R Principle: Description of a Specimen-Specific Finite Element Model Simulating 3-Point-Bending Tests in Mouse Tibiae

**DOI:** 10.3390/bioengineering9080337

**Published:** 2022-07-25

**Authors:** Xiaowei Huang, Andreas K. Nussler, Marie K. Reumann, Peter Augat, Maximilian M. Menger, Ahmed Ghallab, Jan G. Hengstler, Tina Histing, Sabrina Ehnert

**Affiliations:** 1Siegfried Weller Research Institute, BG Unfallklinik Tübingen, Department of Trauma and Reconstructive Surgery, Eberhard Karls University of Tübingen, 72076 Tübingen, Germany; xiaoweih008@gmail.com (X.H.); andreas.nuessler@gmail.com (A.K.N.); mreumann@bgu-tuebingen.de (M.K.R.); mmenger@bgu-tuebingen.de (M.M.M.); thisting@bgu-tuebingen.de (T.H.); 2Department of Orthopedics, The First Affiliated Hospital of Soochow University, Suzhou 215006, China; 3Institute for Biomechanics, Paracelsus Medical University Salzburg, Austria & BG Unfallklinik Murnau, 82418 Murnau, Germany; peter.augat@bgu-murnau.de; 4Department of Toxicology, Leibniz Research Centre for Working Environment and Human Factors (IfADo), 44139 Dortmund, Germany; ghallab@ifado.de (A.G.); hengstler@ifado.de (J.G.H.); 5Department of Forensic Medicine and Toxicology, Faculty of Veterinary Medicine, South Valley University, Qena 83511, Egypt

**Keywords:** finite element analysis, biomechanics, long bone, rodents, validation

## Abstract

Bone mechanical properties are classically determined by biomechanical tests, which normally destroy the bones and disable further histological or molecular analyses. Thus, obtaining biomechanical data from bone usually requires an additional group of animals within the experimental setup. Finite element models (FEMs) may non-invasively and non-destructively simulate mechanical characteristics based on material properties. The present study aimed to establish and validate an FEM to predict the mechanical properties of mice tibiae. The FEM was established based on µCT (micro-Computed Tomography) data of 16 mouse tibiae. For validating the FEM, simulated parameters were compared to biomechanical data obtained from 3-point bending tests of the identical bones. The simulated and the measured parameters correlated well for bending stiffness (R^2^ = 0.9104, *p* < 0.0001) and yield displacement (R^2^ = 0.9003, *p* < 0.0001). The FEM has the advantage that it preserves the bones’ integrity, which can then be used for other analytical methods. By eliminating the need for an additional group of animals for biomechanical tests, the established FEM can contribute to reducing the number of research animals in studies focusing on bone biomechanics. This is especially true when in vivo µCT data can be utilized where multiple bone scans can be performed with the same animal at different time points. Thus, by partially replacing biomechanical experiments, FEM simulations may reduce the overall number of animals required for an experimental setup investigating bone biomechanics, which supports the 3R (replace, reduce, and refine) principle.

## 1. Introduction

Nowadays, worldwide, more than 100 million animals are sacrificed for research every year [1]. In the European Union, except for Portugal, the total number of procedures involving animals in 2016 was 10.5 million—of which almost 70% were used for basic and translational research [2]. The extensive use of experimental animals to investigate physiological mechanisms of human diseases is still controversial in the scientific community and the public. Rodents have been widely accepted as the main source of laboratory animals, of which rats and mice account for more than half of the animals used in bone research [3]. Because of their ease of manipulation, mice became more and more popular as orthopedic models and are nowadays most commonly used to study fracture healing and osteoporosis [4]. Different mouse strains are used to evaluate gene function and adaptive responses to drug treatment or mechanical load [5]. As the exclusive load-bearing structure, the mechanical properties of bones are the most direct parameters that characterize their functionality. These parameters are classically determined by biomechanical tests, e.g., compression, torsion, and 3- or 4-point bending. These biomechanical tests normally destroy the bones, which then mostly cannot be used anymore for further histological or molecular analyses. In long bones, in vivo loads usually include bending, torsion, and compression; therefore, most studies use one of these loading modes [5]. Comparing the different biomechanical testing modes, one has to consider that intact long bones most rapidly fail in a brittle manner during torsion tests. As bending is of higher biological relevance for long bones than unidirectional compression, whole-bone mechanical tests are most often performed in a bending mode [5].

With the rapid development of computers, finite element analysis has evolved to simulate material characteristics not only with regular but also with irregular geometric shapes and various boundary conditions. This makes finite element analysis a powerful tool to predict bone mechanical properties in a non-invasive, non-destructive manner. Thus, using finite element models (FEM), the mechanical properties of bones can be simulated and analyzed based on material characteristics [6]. As one of the first, Pistoia et al. [7] used high-resolution three-dimensional peripheral quantitative computed tomography images of human cadaver distal radii and established a finite element model (FEM), which contained more than two million elements and required about 50 h for calculation. However, it was found that the calculated results using the FEM better predicted the actual measured compression results (R^2^ = 0.75, *p* < 0.001) than dual-energy X-ray absorptiometry (DXA/R^2^ = 0.48, *p* < 0.001) or bone morphology measurements (R^2^ = 0.57, *p* < 0.001). A few years later, Buckley et al. [8] used peripheral quantitative computed tomography to establish an FEM simulating compression of a human vertebral body. The simulation results of this FEM were, again, superior in predicting the experimental results (R^2^ = 0.80, *p* < 0.001) when compared to trabecular or integral bone mineral density (R^2^ = 0.16 and 0.62, respectively). These examples give evidence that FEM simulations could reliably predict the biomechanical situation of dissected bones or even bones within the living body. Therefore, FEM simulations may potentially replace biomechanical experiments and thus contribute to reducing the overall number of animals required for an experimental setup investigating bone biomechanics. Following this assumption, this study aimed to explore the possibility of replacing the 3-point bending test of mouse tibiae in bone research with an FEM simulation within the framework of the 3R (replace, reduce, and refine) principle.

## 2. Materials and Methods

### 2.1. Bone Specimens and µCT (Micro-Computed Tomography) Scanning

Breeding of the C57BL/6 WISP-1 knock-out mice was approved by the local animal welfare committee (Az.: 84-02.04.2018.A284/LANUV, North Rhine-Westphalia, Germany) of the Leibniz Research Centre for the Working Environment and Human Factors (IfADo) and isolation of tibia was registered at the Tübingen district office (DE 08416113021/Az.: 32/9181.21/Mu approval at the 21 February 2018). Each of eight tibiae from wild-type C57BL/6 mice and C57BL/6 WISP-1 knock-out mice, with ages ranging from 8 to 14 weeks, were dissected and cleaned from any remaining muscles, fat, or tissue. In these animals, both the age difference and the knockout of WISP1 may affect bone strength [9,10], which is ideal, as the FEM development and validation require different bone strength. Then, the bones were placed in a 2 mL reaction tube filled with water and scanned under a µCT (µCT 80, SCANCO MEDICAL) with an exposure time of 1000 ms. The rotation pattern was set as 360° of rotation with 360 steps [11]. The resulting scans were cropped and rotated so that all bones were in a straight position, based on a horizontal tibia plateau. Obtained µCT datasets were saved in the format of Digital Imaging and Communications in Medicine (DICOM). Dissected bones were stored at −80 °C until further use.

### 2.2. 3-Point Bending Test for FEM Validation

The mechanical properties of the intact tibiae were evaluated by a standardized three-point bending test. The test was performed on a precision load frame (Zwicki Z2.5 TN, Zwick Roell, Ulm, Germany) at room temperature. The precision loading frame was connected to a Xforce HP force sensor (Zwick Roell/precision of measurements: 2 mV/V/ISO 7500-1). The tibiae were tested with a compressive load on their dorsal side and a tensile load on their ventral side. The bones were positioned with their ventral surface facing downwards in the most stable position possible. The distance between the two supports was 10 mm (Figure 1A). This allowed the stable placement of the tibiae (average length 17.4 mm/Figure 1B) on the supports. The load was applied at a speed of 0.025 mm/s with a preload of 1 N until failure. The testXpert II V3.3 program (Zwick Roell, Ulm, Germany) was used to measure the mechanical properties of the tibiae. Parameters include yield displacement (mm) and stiffness (N/mm). The yield point was defined as the transition point between elastic and plastic deformation and was calculated as the intersection of the load-displacement curve and the 90% slope of the stiffness (Figure 1C).

### 2.3. Establishment of a Finite Element Model to Simulate a 3-Point Bending Setup

Obtained µCT datasets (DICOM) were imported into Mimics software (Materialise NV., Leuven, Belgium) for pre-processing, as shown in Figure 2A,B. The geometry models of the mice tibiae were then reconstructed with a resolution of 34.939 × 34.939 × 34.939 µm^3^. Through thresholding with a fixed threshold of 1100 HU (visually confirmed by the profile line function of the Mimics software) and regional growing function, the segmentation procedure was completed to obtain a 3D geometry assembled by triangulated surface meshes. The reconstructed 3D geometry, in the form of the stereolithography (STL) format, was imported to Geomagic studio software (3D Systems, Rock Hill, SC, USA) to remove free components and for surface smoothing. In addition, the most proximal and distal parts of the bone shell were removed to facilitate volume meshing. The geometry model of a cylinder with a diameter of 2 mm was imported and assembled with the tibia model to simulate the 3-point bending setting.

During the assembly process, only the cylinder can be moved, and the position of the bone model in the universal coordinate system is fixed. In the next step, the bone model was imported to the 3-Matic module of Mimics software, and uniform remeshing with a target triangle edge with a length of 0.08 mm was applied to the 3D geometries (Figure 2C).

The remeshed 3D geometries of the tibia were imported to Hypermesh 14.0 software (Altair Engineering, Troy, MI, USA), and a tetrahedral volume mesh was assigned based on 2D shell meshes. First-order tetrahedral elements were used in the FEM simulations. Only the 3D component was preserved as the inp format and exported to Mimics again for material assignment based on the Hounsfield (Hu) units of the calibration elements from the µCT images (Figure 2D).

The relationship between density and elastic modulus was acquired from Easley et al. [12]. Poisson’s ratio adopts an empirical value of 0.3 [13]. After that, the model was, again, imported to Hypermesh 14.0 software for setting the boundary conditions, as well as defining axial displacement loads. Boundary conditions were created to simulate the 3-point bending test settings. A rigid body was created based on the aforementioned cylinder, with all the nodes fixed at one point through which the load can be applied. A contact pair was created, with the bone surface as the slave surface and the cylinder surface as the master surface. The friction coefficient was set as 0, and the friction type was set as the default. The supporting areas were linked with two points, respectively. The distance between the proximal and distal point was 10 mm. These two points were tied to the chosen points on the bone surface, which belong to the tetrahedral elements. For the proximal point, all degrees of freedom have been constrained, while for the distal point, which can move along the tibia, all degrees of freedom were constrained expect for the direction along the long axis of the tibia (Figure 2E). Based on the three nodes in the cylinder, a local coordinate was defined for applying the displacement load. In the final step, the processed data were then imported into Abaqus 14.0 software (Simulia, Inc., Providence, RI, United States) for calculation and visualization (Figure 2F). The bending stiffness was calculated based on the load-displacement curve derived from the FEM. The obtained results were divided by 0.9 in order to be comparable to the experimental setting, where the yield point was defined as the intersection of the load-displacement curve and the 90% slope of the stiffness.

### 2.4. Statistical Analyses

Yield displacement and stiffness simulated by the FEM were compared to the counterparts measured with the 3-point bending test. The stiffness was defined by the reaction force divided by the maximal displacement of the tibia. The yield displacement was defined by the yield load measured experimentally divided by the bending stiffness calculated by the finite element model. Pearson correlation analyses were employed for model validation. The biomechanical parameters, simulated by the FEM and measured by a 3-point bending test, were compared by a non-parametric paired Wilcoxon test. A *p* < 0.05 was considered statistically significant. Statistical analyses were performed using Graph Pad Prism 8 (San Diego, CA, USA).

## 3. Results

In our study, the average element number of the tibia model was 116,215 ± 2634, and the average node number was 10,942 ± 298. The average bending stiffness calculated by the FEM (114.0 ± 14.8 N/mm) was close to the bending stiffness (115.1 ± 18.2 N/mm, *p* = 0.4637) measured by three-point bending tests, which showed only a −1.0% deviation from the experimental results. With an R^2^ of 0.9104 (*p* < 0.0001), the individual data points for the bending stiffness correlated well between the FEM and 3-point bending test (Figure 3A,B). Similarly, the average yield displacement calculated by the FEM (0.204 ± 0.025 mm) was comparable to the yield displacement (0.197 ± 0.023 mm, *p* = 0.1754) measured by the 3-point bending tests. For the yield displacement, the FEM results showed a 3.6% difference from the experimental test. The individual data points for the yield displacement also correlated well between the FEM and the 3-point bending test (R^2^ = 0.9103, *p* < 0.0001/Figure 3C,D). For the bending stiffness, the non-normalized root-mean-squared-error (RMSE) was 5.10, and the normalized RMSE, with respect to the standard deviation, was 1.06. For the yield displacement, the non-normalized RMSE was 0.011, and the normalized RMSE was 1.400.

## 4. Discussion

Our results show that the simulation based on the FEM can obtain virtual parameters that are very close to the real biomechanical test. Therefore, the FEM can be potentially used to replace biomechanical tests and thus reduce the number of animals that are assigned for biomechanical testing. Being methodologically more straightforward than finite element analysis, classical beam theory has hitherto been used in biomechanics to model the stress behavior of vertebrate long bones. For instance, Arias–Moreno et al. [14] used beam theory to predict the bending stiffness of a rat femur in a typical 3-point bending setting. With an R^2^ of 0.848, the beam theory gave a good approximation of the bending stiffness measured. However, one has to consider that classic beam theory requires an approximation of complex structures (e.g., bones) as slender beams, which might lead to inaccuracies. This might be due to the more complex structure of the tibia in comparison to the femur. The more complex and thus time-consuming FEM requires no such assumptions. This may explain why the FEM presented in this study showed a better correlation with the measured biomechanics than the beam theory model [14], which, when applied to our mice tibiae, resulted in an average −13.4% difference in the yield displacement (*p* = 0.0174) compared to the 3-point bending test.

To simulate real bone mechanics, one of the biggest challenges is the assignment of the bone material properties to mimic the heterogeneity of the bone tissue material. It was found that heterogeneous specimen-specific FEM with a non-uniform material property distribution can improve the accuracy of the bone FEM compared with the homogeneous material assignment [15]. Comparing the inhomogeneous orthotropic and isotropic material assignment in different bone FEMs revealed that using the more realistic inhomogeneous orthotropic material assignment may strongly improve the FEM simulations of the mechanical properties for small bone specimens, e.g., pieces of trabecular or cortical bone, although the study conducted by Oliviero et al. [16] found that hexahedral models with a homogeneous material assignment demonstrated the best correlation. This factor may be omitted for global FEM simulating mechanical properties of intact long bones [17], as in our study. One strategy for assigning material properties to such a global FEM is considering the μCT gray value distribution. In the present study, the local material property assignment has been achieved by converting grayscale values into the material properties of the bone. Therefore, it can be inferred that the actual mouse tibia three-point bending tests can be partially replaced by the FEM analyses, which are based on the µCT data with the literature’s suggested equations. However, it should be noted that the density-elasticity equation was taken from a study in rats [12], which may be slightly different from that of mice. Replacing 3-point bending tests with the non-destructive FEM eliminates the need for an additional group of experimental animals for biomechanical analyses; thus, the FEM may not only reduce the number of experimental animals but also save time and costs by making the most of the µCT data.

Currently, in vivo µCT scanning is becoming more and more popular for non-invasively characterizing bone morphological changes [14]. For example, experimental protocols for studying osteoporosis or analyzing the function of genes on bone quality often consider µCT scans to characterize the morphological characteristics of cortical and cancellous bone. In such a setting, the use of in vivo µCT scans reduces the number of experimental animals compared to the conventional ex vivo µCT scans, as the animals do not need to be sacrificed for each experimental time-point. Presuming an adequate resolution of the in vivo µCT scans, the same would hold for the established FEM. For example, by using in vivo µCT scans, the FEM could be used to identify experimental time-points for further analyses without the need to sacrifice the animals. This way, the FEM could contribute to reducing the total amount of experimental animals within a study protocol.

The linear-elastic nature of the FEM bares limitations. While the bending stiffness can be well simulated using our FEM, the yield displacement requires experimental data. Per the definition, the yield point is the point of transition point between elastic and plastic deformation; thus, reaching the yield point in 3-point bending tests implies damage to the bone structure. Considering that 3-point bending tests are non-destructive when limited to the linear-elastic range (before the yield point), the here presented simple FEM based on ex vivo μCT data is equally good in determining bending stiffness as the biomechanical measurement. However, the option that the FEM utilizes in vivo μCT data shows great potential in reducing the number of mice needed in an experimental setup, as the bending stiffness can be monitored from the same mouse at different time points. Thus, simulating the bending stiffness of the same mouse at different time points would not only reduce the number of mice needed for different time points for ex vivo biomechanical tests but also increase the comparability of the mechanical properties among different time points.

## 5. Conclusions

In conclusion, the FEM based on µCT data of mouse tibiae is well-established with an acceptable deviation from the experimental measures. It can be used to simulate the mechanical behavior (linear-elastic range) of the mouse tibia and predict bending stiffness under a 3-point bending test. Thus, this µCT scanning-based FEM represents a very attractive alternative to the silico method, which can contribute to the 3R principles of the European Union by reducing the number of animals required for bone biomechanical analyses while reducing the experimental time and costs, as well as respecting animal welfare.

## Figures and Tables

**Figure 1 bioengineering-09-00337-f001:**
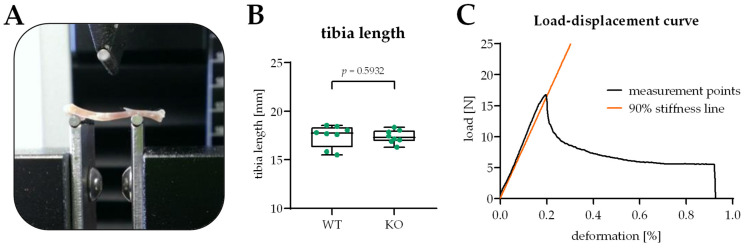
Setup of the 3-point bending test, including an exemplary graphical analysis. (**A**) Photograph of the experimental setup for the 3-point bending test using the Zwicki Z2.5 TN (Zwick Roell, Ulm, Germany) material testing machine. (**B**) Distribution of the length of the tibiae used in this study. (**C**) The exemplary load-displacement curve for the calculation of the yield displacement (mm) and stiffness (N/mm).

**Figure 2 bioengineering-09-00337-f002:**
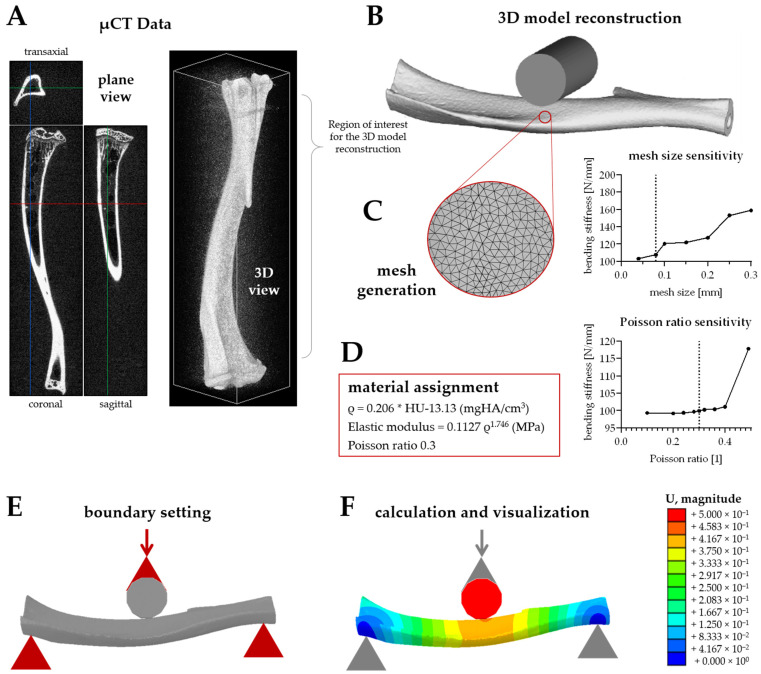
Workflow of the establishment of the finite element model (FEM) to simulate a 3-point bending test in mice tibiae. (**A**) The established FEM is based on µCT data from mice tibiae. (**B**) The reconstructed mice tibiae were surface smoothed, and the region of interest was defined. (**C**) The 3D model was uniformly remeshed before (**D**) the material assignment was performed based on the Hounsfield (Hu) units of the calibration elements from the µCT images. (**C**) Mesh size sensitivity and (**D**) Poisson ratio sensitivity are displayed. (**E**) Then, the boundary setting, 1 loading (from the top) and 2 constrained (from the bottom) points were defined, (**F**) and the FEM simulation was run and visualized. U magnitude represents the displacement distribution after loading.

**Figure 3 bioengineering-09-00337-f003:**
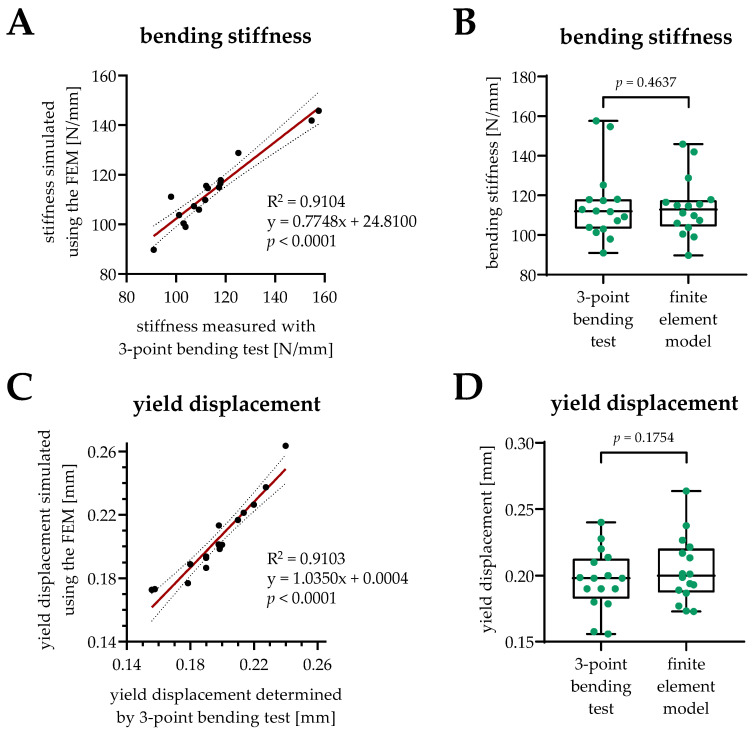
Correlation of the biomechanical parameters obtained by the simulation with the finite element model (FEM) with the biomechanical parameters measured with 3-point bending tests. (**A**,**C**) Bending stiffness [N/mm] and yield displacement [mm] obtained from the biomechanical measurement with the 3-point bending test are displayed on the x-axes, while corresponding parameters from the FEM simulation are displayed on the y-axes. The resulting 95% confidence bands, the correlation coefficient (R^2^), the equation of the best fit line, and the *p*-value are displayed in each graph. (**B**,**D**) Comparison of the measured and simulated biomechanical parameters displayed no significant differences with the Wilcoxon test.

## Data Availability

The datasets generated during and/or analyzed during the current study are available from the corresponding author on reasonable request.

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
