# Peer review of "Contribution to the 3R Principle: Description of a Specimen-Specific Finite Element Model Simulating 3-Point-Bending Tests in Mouse Tibiae"

_bioengineering, 2022, doi:10.3390/bioengineering9080337_

Round 1

Reviewer 1 Report

General comments

The manuscript presents a new computational pipeline for quantifying whole mouse tibia mechanics. The motivation is to partially replace the need for destructive experimental mechanical testing. Overall, the study is promising. However, it is undermined by the presentation of methods, results and discussion. These must be improved in order to be recommended for publication.

Specific comments

Materials and Methods

1. Animal experiments need ethical approval. The text states that a local welfare committee approved it. However, the authors are affiliated to multiple institutions, so it is not clear where "local" is.

2. The microCT image is a key input to the finite-element (FE) model. However, the process of obtaining it is not adequately described. For example, how are the bones held in place during imaging, whether and how the bones are kept hydrated, etc.; and in general, is there an established protocol (then a reference is needed), or was a protocol developed for this study?

3. How is the threshold identified to separate bone/non-bone regions? Is the threshold specific to each mouse, or was a fixed threshold used?

4. Which order of tetrahedral elements were used in the FE simulations? What was the average size of the elements? Were the size varied through the volume (e.g. using sizing functions) or were they more or less uniform? How many elements were there in the typical model?

5. Line 107: should read "... 3-Matic module of Mimics ..."

6. Details of contact between the cylinder and bone should be given.

7. How are the coordinate system axes defined in FE?

8. Specification of boundary conditions is unclear. What is the initial distance between the proximal and distal points? What is the procedure for locating these points on the bone surface? Note that if these points are nodes belonging to tetrahedral (i.e. continuum) elements, then they possess only 3 (not 6) degrees of freedom.

9. The order of the two sentences on the last paragraph of p. 3 appear to be interchanged. Surely the FE analysis was performed before the bending stiffness could be computed.

10. In Figure 1, please give details of U magnitude, and consider whether the number of significant digits used in the legend is sensible (based on mesh independence and uncertainty in model input). The Elastic modulus expression (1D) should have units spelled as MPa.

11. The two support points appears to be vertically separated prior to deformation. Why?

12. Line 130: Please correct "uniformly"

13. Line 132: Please correct "constrained"

14. In general, details of the experiment (§2.3) – which is the ground truth – should be presented before those of image processing and FE simulations (§2.2). As well, for each aspect of the FE model (see comments 6–8, 10 and 11 above), please explain how it relates to the experiment.

15. Lines 137–38, please replace the odd terminology "frontside", "backside", with established technical terms.

16. It would be useful to have a picture of the experimental set-up showing the supports, placement of load-cell, etc. As well, a table with distribution of tibia lengths should be provided.

17. In the mouse, the tibia and fibula are fused. It is not mentioned in the methods whether the fibulae were removed prior to testing.

18. The mouse tibia is very delicate and also quite irregular in shape. As such, it can present a number of challenges in experimentally testing it, e.g. identifying its axis, ensuring that it is stable during loading, etc. In contrast, the details provided in section 2.3 are very sparse.

19. Unclear how yield is identified from experimental load-displacement curve.

20. What is the frequency of load/displacement data acquisition. What is the precision of the load cell and displacement head?

21. Line 145: unclear how yield is detected in FE, which defines only an elastic material response.

22. Since for each mouse bone, both an FE and a mechanical test was conducted, is it not more meaningful to perform a Mann-Whitney-Wicoxon (paired samples) test?

Results

23. Need to show at least a representative load-displacement curve from the experiments, and locate the yield point on the curve. Showing the full spread of experimental data is even better.

24. Please also state the root-mean-squared-error (both normalised and non-normalised with respect to the standard deviation) in predicting the experimental results using FE. This will allow someone applying only the FE pipeline to estimate the uncertainty in their approach.

25. In lines 154 and 159, it is unclear what are the quantities (average, SD, abs-max, etc.) with respect to which the percentage values are expressed. Suggest using an equation to express this precisely.

Discussion

26. The density–elasticity equation is taken from a study (ref. 9) in rats, which are different from mice. This needs to be discussed.

27. Line 193: It is not always the case that using heterogeneous modulus will provide more accurate results, but needs to be demonstrated for various loading conditions. Note that the study [12] is from rabbits, a species that is much larger in scale than mice; whereas a study done in mice (doi:10.1007/s10237-021-01422-y) reached a very different conclusion.

Author Response

We would like tot thank the reviewer for this detailed and thorogh revision of our manuscript. Very good questions were raised. Please find our detailed answered attached.

Reviewer 2 Report

Scope: The content of the manuscript is strongly related with the scope of the journal.

Language:

The authors use some abbreviations without defining, most typical are μCT and 3R principle. These should be defined at first occurence in the text.

The use of English is overall correct, wih a few fragmentations like, line 184 "This 184 may explain why the here presented FEM showed better correlation with the measured 185 biomechanics ..." . This broken sentence must be corrected.

In line 173 the tense should have a common reference point. 

Novelty : 

The paper compares bending stiffness and yield displacement of intact tibiae with those obtained from a FEM with material properties assigned from literature suggested values.  The results show that there is good correlation, thus it can be concluded that material properties for the mice bone can be used in FEM models avoiding mechanical tests each time. 

The the tests and results are original additions to the literature and significant.

Quality and scientific soundness :

The text lacks some important information as to the FEM. 

1) Is this a linear or nonlinear analysis (small strains, small displacements ?) ? Does material properties depend on level of strain ? Does the material have a yield stress level, if so what yield criteria is used in the analysis ?

2) Give some info about the finite elements used in the model, number of nodes etc. Give a figure describing the elements used in the model.

3) When referencing always mention the first work. Elastic modulus eqquation is taken from reference (9), but reference 9 actually takes it from Currey (1988). Please refer to the first reference.

4) Could you please discuss why you take Poisson's ratio as 0.3 ? Beyond the literature, do you have any reasons that support this value as appropriate? Did you try other values for Poisson's ratio ? Should the authors present a sensitivity analysis of stiffness and yieÅŸd displ. as to the variation of Poisson's ratio, then the scientific contribution of the paper would be better. 

5) Would you please define stiffness and yield displacement ? 

6) can you summarise your results in a few tables, graphs ? For instance, can you give a load displacement curve indicating the yield displacement ?

7) How do you quantify (compute) stifness and yield displacement ? 

Regarding the experiments, how many specimen were used in the tests ? 

Most critical is the claim that "FEM can replace experiments..." . On the contrary, a FEM model requires material properties to be entered, as the authors did, using modulus of elasticity and poissons ratio from literature. We make experiments to determine material properties. However, the authors can claim that the literature suggested equations are reliable anough to create (preliminary) FEM  models avoiding experiments. 

The paper should be improved in line with the suggestions above then it will appeal to the interest of the readers in the field of biomechanics. 

Author Response

We would like tot thank the reviewer for his/her thorogh revision of our manuscript. Please find our detailed answered attached.

Reviewer 3 Report

This paper discussed about specimen specific finite element simulation of 3-point-bending tests of mouse tibiae. The result of this study is interesting to the readers. The manuscript is also well-written. However, some additional information should be added to improve the quality of the manuscript. My comments are listed below.

1. Mesh size is believed to give significant effect on the bending stiffness and yield displacement. How did you decide the mesh size? Please provide more detail explanation.

2. Could you give the roughly dimensions of the tibiae used in the study?

3. How did you define “yield point” in the FE simulations?

4. Figure 2 shows huge variations in the results. Could you provide a discussion regarding the reasons of these variations? 

Author Response

(The authors gave the same response as above.)

Round 2

Reviewer 1 Report

I thank the authors for revising the manuscript and addressing all the comments in full. I have some further comments as follows.

1. Different definitions of yield displacement are used in experiments vs FE simulations.

In the experiments (red line in Fig 1C), it appears that the yield displacement (let's call it d_exp) would satisfy the relationship d_exp = F / (0.9*k_exp) where F is the experimentally determined load at yield and k_exp is the experimentally determined stiffness (in the elastic regime prior to yield).

In contrast, in the FE simulations, the yield displacement is defined as d_FE = F / k_FE, where "_FE" denotes the source of the estimate, and F - as before - is the experimentally determined load at yield. The definitions of k_FE and k_exp are identical, although the source differs.

Could the authors discuss the need for the different definitions of the yield displacement? Does this difference not already explain the finding that d_FE < d_exp? Would using the same definition lead to a better match between simulations and experiments?

2. Since F (from experiments) is needed to determine d_FE, clearly the FE pipeline cannot replace the experiment. The authors should discuss this limitation in the context of 3Rs. Note that if only bone stiffness needs to be determined, one does not need to break the bone i.e. the experiment can be performed non-destructively and so the value of FE - in being a non-destructive option - is relatively lower. This should be discussed.

Author Response

We would like to thank the reviewer for raising these additional points. We recalculated the data according to the reviewer's suggestion, which strongly improved the comparability of the FE Model with the experimental data.

In addition, the limitations of the FEM were further discussed.

Reviewer 2 Report

The authors have improved all the manuscript regarding all the corrections requested in the first round. No furteher improvements are necessary.

The paper is ready to be published as is in the revised form. 

Author Response

We would like to thank the reviewer for his/her kind estimate on our revised manuscript.

Reviewer 3 Report

Thank you for the answers. The manuscript can be accepted for publication.

Author Response

(The authors gave the same response as above.)
